# Folate deficiency among women of reproductive age in Ethiopia: A systematic review and meta-analysis

Berhe Gebremichael [1]*, Hirbo Shore Roba[1,2], Alemeshet Getachew[1], Dejene Tesfaye[3], Haftu Asmerom[4]

1 School of Public Health, College of Health and Medical Sciences, Haramaya University, Harar, Ethiopia,
2 School of Health and Medical Sciences, University of Southern Queensland, Queensland, Australia,
3 Department of Psychiatry, School of Nursing and Midwifery, College of Health and Medical Sciences, Haramaya University, Harar, Ethiopia, 4 School of Medical Laboratory Science, College of Health and Medical Sciences, Haramaya University, Harar, Ethiopia

* berhegere09@gmail.com

## Abstract

### Background

Folate deficiency (FD) can cause adverse health outcomes of public health significance. Although FD is a significant micronutrient deficiency in Ethiopia, concrete evidence is limited. Therefore, this systematic review and meta-analysis was designed to estimate the pooled prevalence of FD among women of reproductive age (WRA).

### Methods

A systematic literature search was performed using MEDLINE, Embase, CINAHL, Google Scholar, African Journals Online (AJOL), The Vitamin and Mineral Nutrition Information System (VMNIS) of the World Health Organization (WHO), Global Health Data Exchange (GHDx), and institutional repositories of major universities and research centers. Additionally, we scanned the reference lists of relevant articles. Two authors independently selected the studies, extracted the data, and the study risk of bias. Heterogeneity was assessed using the $I^2$ statistic. We used a random-effects model to estimate the pooled mean serum/plasma folate and the pooled prevalence of FD. Begg's and Egger's tests were used to check publication bias.

### Results

Ten studies—nine cross-sectional and one case-control—with a total of 5,623 WRA were included in the systematic review and meta-analysis. Four (WRA = 1,619) and eight (WRA = 5,196) cross-sectional studies were used to estimate the pooled mean serum/plasma folate and prevalence of FD, respectively. The pooled mean serum/plasma folate concentration estimate was 7.14 ng/ml (95% CI: 5.73, 8.54), and the pooled prevalence of FD was estimated to be 20.80% (95% CI: 11.29, 32.27). In addition the meta-regression analysis

**Data Availability Statement:** All relevant data are within the paper and its Supporting information files.

**Funding:** The author(s) received no specific funding for this work.

**Competing interests:** The authors have declared that no competing interests exist.

**Abbreviations: AJOL**, African Journals Online; **CI**, Confidence Interval; **CINAHL**, Cumulative Index to Nursing and Allied Health Literature; **FD**, Folate Deficiency; **GHDx**, Global Health Data Exchange; **IQR**, Inter-quartile Range; **NTD**, Neural Tube Defect; **RBC**, Red Blood Cell; **SD**, Standard Deviation; **VMNIS**, Vitamin and Mineral Nutrition Information System; **WHO**, World Health Organization; **WRA**, Women of Reproductive Age.

showed that the sampling technique was significantly associated with mean serum/plasma folate concentration.

## Conclusions

FD is a significant public health issue among WRA in Ethiopia. Therefore, the public health strategies of the country should focus on promoting the consumption of folate-rich foods, strengthening the coverage of folic acid supplementation and its adherence, and swift translation of the mandatory folic acid fortification into action.

## Systematic review registration

PROSPERO 2022—CRD42022306266.

## Introduction

Folate is a naturally occurring essential vitamin mainly found in green leafy vegetables and legumes [1]. It is necessary for deoxyribonucleic acid (DNA) replication and normal cell formation and growth [2]. Therefore, folate deficiency (FD) reduces thymidylic acid and increases homocysteine in the body, resulting in several health risks [3].

Folate deficiency is caused primarily by inadequate dietary intake [4, 5] and partly by medical and physiologic conditions that increase the need or excretion of folate [5–8]. Different study findings show that iron deficiency anemia [4, 9], knowledge of folate-rich foods [4, 10], and folic acid supplementation [11] are significantly associated with FD. Folate intake is suboptimal in the diets of many WRA and exacerbated by overcooking foods and poor bioavailability [8], estimated to be from 50% to 82% [12, 13]. In several developed countries, fortifying grains with folic acid has increased folate intake [14]. However, in Ethiopia, the unavailability of fortified foods further increases the risk of folate insufficiency, particularly for vulnerable populations such as WRA [4, 15].

A definition for what constitutes a public health problem for folate deficiency is not well established [16] due to limited relevant population-based data [17]. However, a 5% and above prevalence generally represents a public health problem [16–19]. Therefore, folate deficiency is considered a severe public health issue, especially among disadvantaged groups in developing countries, including Ethiopia [20, 21]; some of the most affected groups by FD include WRA [5, 7, 17].

According to a global systematic review based on the available evidence of folate status in WRA, the prevalence of FD was greater than 20% in many low-income countries, which is far above the general threshold for public health concerns. However, in high-income countries, the prevalence was less than 5%. In this review the prevalence of folate insufficiency was more than 40% in most countries [22]. Among different African countries, the prevalence of FD in pregnant women fluctuated from 0.8% in Kenya to 86.1% in Côte d'Ivoire [9, 23–26]. Similarly, different study findings in Ethiopia show the prevalence of FD in WRA ranging from 1.9% to 46% [4, 10, 27–32]. However, in most of these studies, the prevalence of FD was above 5%—indicating its public health significance in the country.

The consequences of FD among WRA include megaloblastic anemia [8, 33–35] and neural tube defects (NTDs) [36–39]. Maternal anemia is highly prevalent in Ethiopia [40]; however, reports showing FD as the etiology of anemia are unavailable. Folate deficiency is a significant

risk factor for neural tube defects (NTDs), which affects more than 300,000 births worldwide and 65 per 10,000 births in Ethiopia [36–39]. As countries progress in reducing child mortality from infectious diseases, congenital disabilities become a more significant cause of under-five mortality in many countries [41]. Other consequences of FD include abortion [33–35], pre-term birth [33], and hyperhomocysteinemia, which is a risk factor for metabolic and cardio-vascular diseases [3, 34, 35, 42].

Folate deficiency among WRA is one of the significant public health issues in Ethiopia [43]. Therefore, to address this problem, the Ethiopian Government adopted the global targeted iron and folic acid supplementation for pregnant women during their antenatal care visits to reduce the prevalence of anemia in WRA and children under five [44]. Additionally, in 2022, the country endorsed the mandatory fortification of edible oil and wheat flour with folic acid, an effective intervention strategy to overcome the burden of FD [45]. However, to our knowl-edge, no systematic review and meta-analysis study has addressed FD among WRA in Ethiopia to provide high-level evidence for policymakers to track the progress, evaluate the impact of existing programs and design further context-specific interventions. Therefore, this systematic review and meta-analysis was designed to address the gap and estimate the exact prevalence of FD among WRA in Ethiopia.

## Methods and materials

### Registration

This systematic review and meta-analysis was performed, according to the protocol registered in the International Prospective Register of Systematic Reviews (PROSPERO) on 22 February 2022, with registration ID: CRD42022306266.

### Search strategy

Studies were identified by searching electronic databases, institutional and organizational repositories, and websites. Additionally, the reference lists of key articles were examined to retrieve additional related studies. From the electronic databases, we searched MEDLINE (via PubMed), Embase (via Ovid), CINAHL (via EBSCOhost), African Journals Online (AJOL), and Google Scholar. The Vitamin and Mineral Nutrition Information System (VMNIS) of the World Health Organization (WHO), Global Health Data Exchange (GHDx), institutional repositories of major universities (including Addis Ababa University, Jimma University, Hawassa University, Haramaya University, Arbaminch University, University of Gondar, Bahir Dar University, and Mekelle University), and the institutional repositories of research centers (including the Ethiopian Public Health Institute (EPHI), Ethiopian Health and Nutri-tion Research Institute (EHNRI) and Ethiopian Nutrition Institute (ENI)) were searched for reports and unpublished articles.

The search was first conducted on 28 February 2022, and updated on 31 May 2022. Two authors (BG and HSR) performed the search activities independently. The following search terms were used to find all relevant studies in the databases and other sources: "prevalence", "magnitude", "status", "level", "folate", "folic acid", "micronutrient", "deficiency" and "Ethiopia". The search terms were used separately and in combination using the Boolean operators 'OR' and 'AND' (S1 Text). The PRISMA guideline for systematic review [46] was used to report the search results.

## Eligibility criteria

The studies which fulfilled the following criteria were included: all observational study designs (cross-sectional, case-control, and cohort studies) which reported mean or median serum or plasma folate, or prevalence of FD, published in the English language from 2004 to 2022, full-text articles, conducted among WRA (15–49 years old) in Ethiopia, studies with a response rate greater than 80%, reporting quality assurance methods, and quality assessment score better than 50%. For any studies from the same survey, we selected and included the study that reported the desired outcomes and the extracted variables clearly. Studies were excluded if they were found to have a poor quality score as per the stated criteria, were review articles, qualitative studies, abstracts, or failed to determine the desired outcomes (i.e., mean or median serum or plasma folate or FD).

The outcome variable was 'folate deficiency' and was defined according to the WHO recommendation as follows: (1) serum or plasma folate <3 ng/mL or red blood cells (RBC) folate <100 ng/mL (using macrocytic anaemia as a haematological indicator); (2) serum or plasma folate <4 ng/mL or RBC folate <151 ng/mL (using homocysteine concentration as a metabolic indicator). Folate insufficiency (level of folate below which is a risk for NTD) was defined as RBC folate < 400 ng/mL [47].

## Study risk of bias and quality assessment

We assessed the risk of bias and quality of the included studies using the Joanna Briggs Institute (JBI) Critical Appraisal Checklist for Studies Reporting Prevalence Data [48]. The checklist contains nine items: (1) appropriateness of the sample frame to address the target population, (2) appropriateness of sampling procedure/technique, (3) adequacy of sample size, (4) detail description of study subjects and setting, (5) sufficiency of data analysis coverage of the identified sample, (6) validity of methods used for the identification of the condition, (7) measurement of the condition in a standard and reliable way for all WRA, (8) appropriateness of statistical analysis, and (9) adequacy of response rate and appropriateness of management of low response rate.

Two review authors (BG and HSR) independently applied the checklist to assess the risk of bias in each included study. If there were any discrepancies between the two review authors, first, the two authors discussed the issue trying to reach a consensus, with a third review author (AG, DT or HA) acting as an arbiter if necessary. Finally, studies with a quality assessment score of greater than 50% were included in the systematic review; fortunately, all the assessed studies scored above 60% (S1 Table).

## Data collection (extraction) process

A structured data extraction format, prepared using Microsoft Excel 2010, was used to extract all the necessary data. Two reviewers (BG and HSR) independently extracted the data from the eligible studies. Disagreements between the two authors were resolved through discussions and consultations with a third review author (AG, DT or HA). For each included study, we extracted the following study characteristics: authors' names and publication date, study year, study setting, study scale/level, study population, sample size, sampling technique, response rate, mean age of WRA, fasting status of WRA during blood sample taking, laboratory assay methods and biomarkers used, cut-off-point used, mean or median serum and/or RBC folate concentration, the prevalence of FD and associated factors of FD.

## Data synthesis and analysis

The meta-analyses were performed using STATA version 16.0 statistical software package. A random-effects model was applied to estimate the pooled mean serum/plasma folate level and the pooled prevalence of FD in WRA [49]. We used the *metan* and *metaprop* STATA commands to estimate the pooled mean serum or plasma folate and the pooled prevalence of FD, respectively. The standard error for mean serum/plasma folate was generated using the following formula: $se = \frac{SD}{\sqrt{n}}$ [50], where se = standard error, SD = Standard Deviation and n = sample size. We generated and used the standard error instead of standard deviation to give more weight to studies with precise estimates and to include the role of sample size in the estimation.

Forest plots were used to present the results of pooled estimates with a 95% confidence interval (CI). The $I^2$ statistic was used to test heterogeneity among studies. To explore the sources of heterogeniety, subgroup analysis was done for the mean serum/plasma folate level by survey year and sampling technique. Additionally, subgroup analysis was done for the prevalence of FD by population group and mean age of WRA. Begg's rank correlation test and Egger's linear regression test were used to check for the publication bias. Leave-one-out sensitivity analysis was performed to show how each individual study affects the overall estimate of the rest of the studie. Finally, meta-regression analyses were carried out to identify parameters associated with FD.

## Ethics statement

All analyses were based on previously done original studies. Therefore, no ethics approval or patient/participant consent was required.

## Results

### Study selection process

Initially, 1600 studies were identified from the electronic searches. Of these, 1560 were published articles, while 40 were theses and organizational reports. After removing 97 duplicates, 1503 records were screened by title and abstract, and 1467 studies were excluded. Next, 36 records were further assessed for risk of bias and eligibility, and 26 studies were excluded (23 did not meet the eligibility criteria, and three had duplicated content). Finally, ten studies were included in the systematic review and meta-analysis (Fig 1).

### Characteristics of included studies

This systematic review and meta-analysis included 10 studies [4, 10, 27–32, 51, 52] with a total of 5,623 WRA, which were conducted between 2004 and 2022. Among the studies, nine [4, 10, 27–32, 52] were cross-sectional studies, and one was a case-control [51]. Regarding the study setting, six of the studies [4, 27, 28, 30, 31, 52] were done at the community level, while the other four [10, 29, 32, 51] were institutional-based. Three studies [4, 28, 30] were conducted at a national level, whereas the other seven were conducted at the sub-national/local level—three in Oromia Region [29, 32, 52], one in Amhara Region [31], one in Sidama Region [27], and two in Addis Ababa City Administration [10, 51].

Six studies [10, 27, 29, 32, 51, 52] were conducted among pregnant women, and two were [28, 30] done among non-pregnant/non-lactating women. However, two studies [4, 31] were performed generally in WRA without specifying the physiological status of the women. Five studies [4, 28, 30, 31, 52] applied probability sampling techniques, while the other five [10, 27, 29, 32, 51] used non-probability sampling. The sample size of the studies ranged from 99 [27]

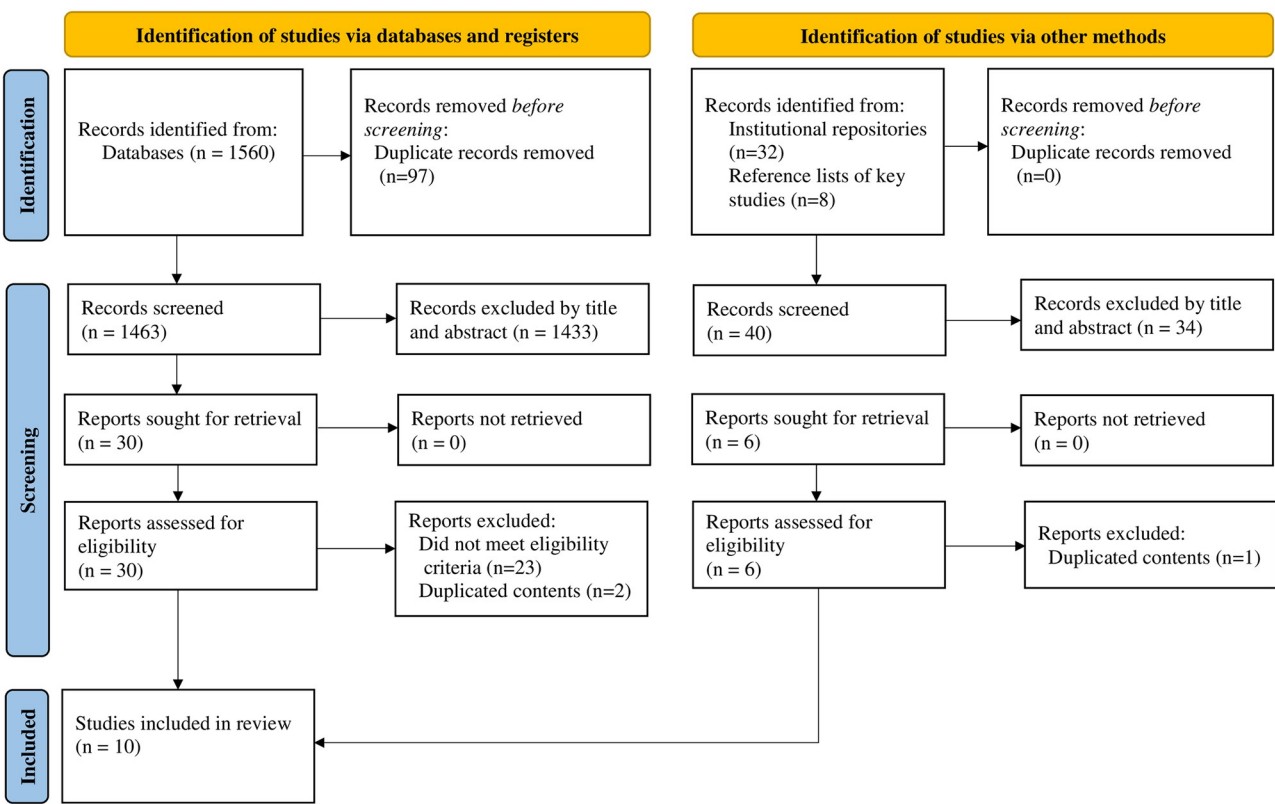

**Fig 1. PRISMA flow diagram showing the selection process of studies for the systematic review and meta-analysis on folate deficiency among women of reproductive age in Ethiopia, 2022.**

to 1647 [30], with response rates ranging from 89.5% [31] to 100% [4, 28, 29, 32, 51]. Nine of the studies [4, 10, 27, 29–32, 51, 52] reported mean (+ SD) age of the WRA, ranging from 24.5 (+ 5.0) [32] to 33.1 (+ 7.2) [31] years.

Concerning the laboratory methods, three studies [4, 10, 51] used fasting blood samples, one [27] used non-fasting blood samples, and six studies [28–32, 52] did not report the fasting status. Seven studies [4, 10, 27, 29, 32, 51, 52] used one of the protein binding assays to assess folate status, two studies [30, 31] used microbiological assays, and one study [28] did not report the type of assay used. Additionally, eight studies [4, 27–29, 31, 32, 51, 52] reported serum/plasma folate as a biomarker, one reported RBC folate [10], and one study reported both serum and RBC folate [30].

Out of those included, six of the cross-sectional studies [4, 27, 29, 31, 52] reported mean (+ SD) serum/plasma folate concentration, ranging from 4.5 ng/ml (+ 0.1) in Oromia Region [52] to 12.6 ng/ml (SD not reported) in Amhara Region [31]. The single case-control study [51] also reported mean (+ SD) serum/plasma folate concentration for the cases (4.4 ng/ml (+ 1.8)), and controls (8.0 ng/ml (+ 3.0)), separately. On the other hand, one cross-sectional study [32] reported median (IQR) serum folate concentration as 4.2 ng/ml (2.3) and the case-control study [51] reported 4.8 ng/ml (3.7) for the cases and 8.9 ng/ml (5.3) for the controls. Whereas, one study [10] reported median (IQR) RBC folate concentration as 585 ng/ml (IQR not reported), and another one study [30] reported both median (IQR) serum and RBC folate concentrations as 5.0 ng/ml (4.5) and 255.5 ng/ml (174), respectively.

Eight of the cross-sectional studies [4, 27–32, 52] reported FD fluctuating from 1.9% [31] in Amhara Region to 46.0% [4] in a national study. Three of these studies [4, 29, 31] also reported possible or marginal FD, varying from 2.6% [31] to 21.2% [4]. On the other hand, the case-control study [51] merged the FD and possible FD, and found 57% for the cases and 33.5% for the controls. However, one cross-sectional study [10] reported folate insufficiency instead of FD.

From all studies, two [4, 52] described four factors associated with FD, while another study [10] found one associated with folate insufficiency. Accordingly, poor vegetable and grain intake [4], poor knowledge on folate-rich foods [52], iron deficiency anemia [4, 52], and no iron-folic acid supplementation [52] were identified as the significant predictors of FD. Similarly, poor vegetable intake [10] was significantly associated with folate insufficiency (Table 1).

## Risk of bias and quality assessment

The risk of bias and quality of the studies was assessed by the Joanna Briggs Institute (JBI) Critical Appraisal Checklist for Studies Reporting Prevalence Data [48], a 9-item based checklist, indicated in the methods and materials section. The overall quality assessment score, with a potential total of 9 points, ranged from 6 (66.7%) to 9 (100%) points. In all (100%) of the included studies, the study subjects and settings were described in detail; valid methods were used for the identification of the condition; the data analysis was performed with sufficient coverage of the identified samples and response rates were adequate. However, the sampling of WRA was inappropriate in 5 (50%) of the studies and sample size was inadequate in 6 (60%) studies (S1 Table).

## Mean serum/plasma folate concentration

Four cross-sectional studies totaling 1,619 WRA were included in the meta-analysis to estimate the pooled mean serum/plasma folate level. Three of the studies [4, 27, 29] were published articles, and one was a master thesis [52]. Two of the studies were from Oromia Region [29, 52], one was from Sidama Region [27], and one [4] was a national study. Concerning the study setting, three studies [4, 27, 52] were community-based whereas one [29] was an institution-based study. The studies were conducted in 2004 [27], 2005 [4], 2015 [29] and 2021 [52]. Three studies [27, 29, 52] were conducted in pregnant women, while one [4] was in WRA. All four studies used protein binding assay to analyze the serum/plasma folate level. Two studies [4, 52] used probability sampling and the other two [27, 29] used non-probability sampling. The sample size of the studies ranged from 99 [27] to 970 [4], with a response rate ranging from 94.5% [27] to 100% [4, 29]. The mean (+ SD) serum/plasma folate concentration varies from 4.5 ng/ml (+ 0.1) [52] to 11.5 ng/ml (+ 5.6) [27] (Table 1).

The meta-analysis revealed that the pooled estimate of the mean serum/plasma folate concentration was 7.14 ng/ml (95% CI: 5.73, 8.54). The analysis revealed considerable between-study heterogeneity ($I^2$ = 99.0%, p = 0.000) (Fig 2). The Egger's test detected significant publication bias (p = 0.000), however, it was not detected in the Begg's test (p = 0.09).

## Prevalence of folate deficiency

The meta-analysis included eight cross-sectional studies conducted from 2004 to 2021, with a total of 5196 WRA, to estimate the pooled prevalence of FD. Four of the studies [4, 27–29] were published articles, three [31, 32, 52] were master theses, and one [30] was an organization survey report. Six of the studies [4, 27, 28, 30, 31, 52] were community-based and two [29, 32] were institutional-based. Of the eight studies, three [4, 28, 30] were done at national level, while five were conducted at the sub-national/local level—three studies were from Oromia

**Table 1. Summary characteristics of the 10 studies included in the systematic review and meta-analysis of folate deficiency among women of reproductive age in Ethiopia, 2022.**

| Study | Study year | Study area | Population | Sampling | Study setting | Fasting status | Lab assay method | Bio-marker | Cutt-off (in ng/ml) | Sample size | Response rate | Mean age (+SD) in years | Mean folate (+SD) in ng/ml | Median folate (IQR) in ng/ml | Marginal FD (%) | FD (%) | Factors associated with FD |
|---|---|---|---|---|---|---|---|---|---|---|---|---|---|---|---|---|---|
| Adela et al., 2018 | 2017 | Addis Ababa | Pregnant women | Non-probability | Institution | Fasting | PBA (ECLIA; Roche Elecsys®) | RBC | <400 | 160 | 98.8 | 26.5 (4) | NR | 585 (NR) | NR | 27.5* | Reported[a] |
| Bekele and Baye, 2019 | 2018 | Amhara | WRA | Probability | Community | NR | MBA (5-methyl-THF calibrator) | Serum | <6.6; <4 | 179 | 89.5 | 33.1 (7.2) | 12.6 (NR) | NR | 2.6 | 1.9 | NR |
| Bromage et al., 2021 | 2019 | National | Non-pregnant/non-lactating | Probability | Community | NR | NR | Serum | <3 | 1604 | 100 | NR | NR | NR | NR | 31.8 | NR |
| Elema et al., 2018 | 2015 | Oromia | Pregnant women | Non-probability | Institution | NR | PBA (ECLIA; Roche Elecsys®) | Serum | <3 | 104 | 100 | 24.6 (5) | 7.6 (3.5) | NR | 10.6 | 17.3 | NR |
| EPHI, 2016 | 2015 | National | Non-pregnant | Probability | Community | NR | MBA (calibrator NR) | RBC | <151 | 1647 | 94.5 | 28.9 (8.8) | NR | 255.5 (174) | NR | 32.0 | NR |
| | | | | | | | | Serum | <4 | 1647 | 94.5 | 28.9 (8.8) | NR | 5.0 (4.5) | NR | 17.3 | NR |
| Gibson et al., 2008 | 2004 | Sidama | Pregnant women | Non-probability | Community | Non-fasting | PBA (RIA; DPC Dual CountTM Solid Phase No Boil) | Plasma | <3 | 99 | 94.9 | 27.8 (4.6) | 11.5 (5.6) | NR | NR | 2.1 | |
| Haidar et al., 2010 | 2005 | National | WRA | Probability | Community | Fasting | PBA (ELISA; Roche Elecsys®) | Plasma | <6.6; <4 | 970 | 100 | 32.6 (12.5) | 5.6 (3.8) | NR | 21.2 | 46 | Reported[b] |
| Kucha et al., 2022** | 2020 | Addis Ababa | Pregnant women | Non-probability | Institution | Fasting | PBA (ELISA; Cloud-Clone Corp. Katy, TX, USA) | Serum | <6 | Cases: 100 | 100 | Cases: 26.8 (5.3) | Cases: 4.4 (1.8) | Cases: 4.8 (3.7) | Cases: 57 | | NR |
| | | | | | | | | | | Controls: 167 | 100 | Controls: 27.2 (4) | Controls: 8 (3) | Controls: 8.9 (5.3) | Controls: 33.5 | | |
| Mebratu and Baye, 2016 | 2015 | Oromia | Pregnant women | Non-probability | Institution | NR | PBA (ECLIA; Roche cobas® c501) | Serum | <3 | 147 | 100 | 24.5 (5) | NR | 4.2 (2.3) | NR | 21.8 | NR |
| Yusuf et al., 2021 | 2021 | Oromia | Pregnant women | Probability | Community | NR | PBA (ECLIA; Roche cobas® e411) | Serum | <4 | 446 | 96.8 | 25.7 (5.2) | 4.5 (0.1) | NR | NR | 49.3 | Reported[c] |

SD: standard deviation

IQR: inter-quartile range

FD: folate deficiency

PBA: protein binding assay

ECLIA: electrochemiluminescence immunoassay

RBC: red blood cell

NR: not reported

WRA: women of reproductive age

MBA: microbiological assay

THF: tetrahydrofolate

EPHI: Ethiopia Public Health Institute

RIA: Radioimmunoassay

ELISA: enzyme-linked immunosorbent assay

a poor vegetable intake was reported as associated factor

b poor vegetable and grain intake, and iron deficiency anemia were reported as associated factors

c poor knowledge on folate rich foods, iron deficiency anemia and iron-folic acid supplementation were reported as associated factors

*This study reported folate insufficiency, not folate deficiency

**This is the only case-control study, and it did not report the possible/marginal FD and FD separately; they were merged

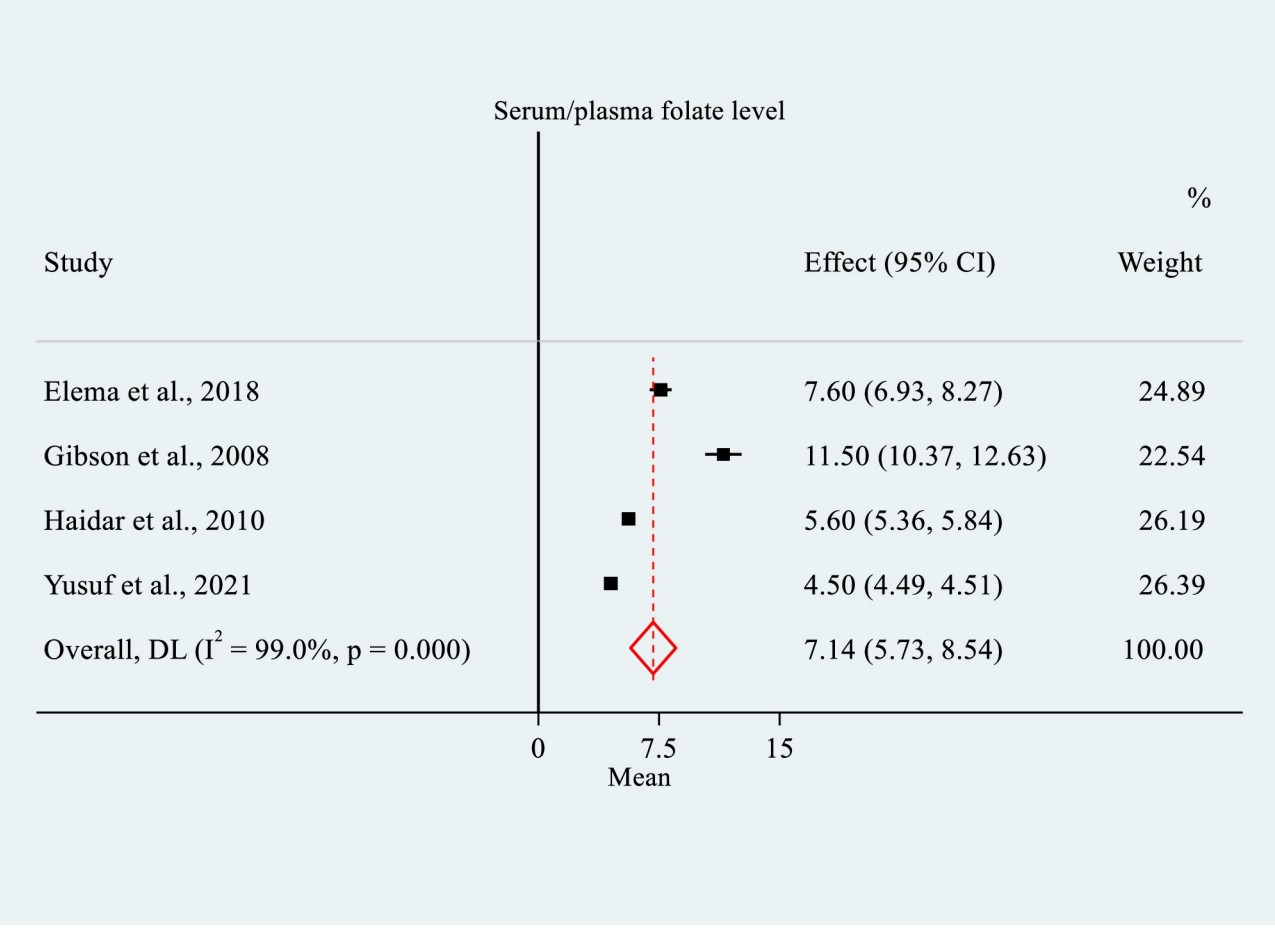

**Fig 2. Forest plot of mean serum/plasma folate level among women of reproductive age in Ethiopia, 2022.** NOTE: Weights are from random-effects model.

Region [29, 32, 52], one from Amhara Region [31], and one from Sidama Region [27]. Pregnant women were the target group in four studies [27, 29, 32, 52], and non-pregnant/non-lactating women were the targets of two studies [28, 30]. Nevertheless, two studies [4, 31] were done generally in WRA without specifying the physiological status of the women. Five studies [4, 27, 29, 32, 52] used protein binding assays, two [30, 31] used microbiological assays, and one [28] did not report the type of laboratory assay. From the studies, five [4, 28, 30, 31, 52] applied probability sampling techniques, while three [27, 29, 32] used non-probability sampling. The sample varied from 99 [27] to 1647 [30], with response rates ranging from 89.5% [31] to 100% [4, 28, 29, 32]. The prevalence of FD was widespread from 1.9% [31] in a study reported from Amhara Region to 46.0% [4] in a national study (Table 1).

In this meta-analysis, the pooled prevalence of FD was estimated to be 20.80% (95% CI: 11.29, 32.27). The meta-analysis indicated marked evidence of heterogeneity among the studies ($I^2$ = 98.71%, p = 0.00) (Fig 3). No significant publication bias was detected using Egger's (p = 0.51) and Begg's (p = 0.17) tests.

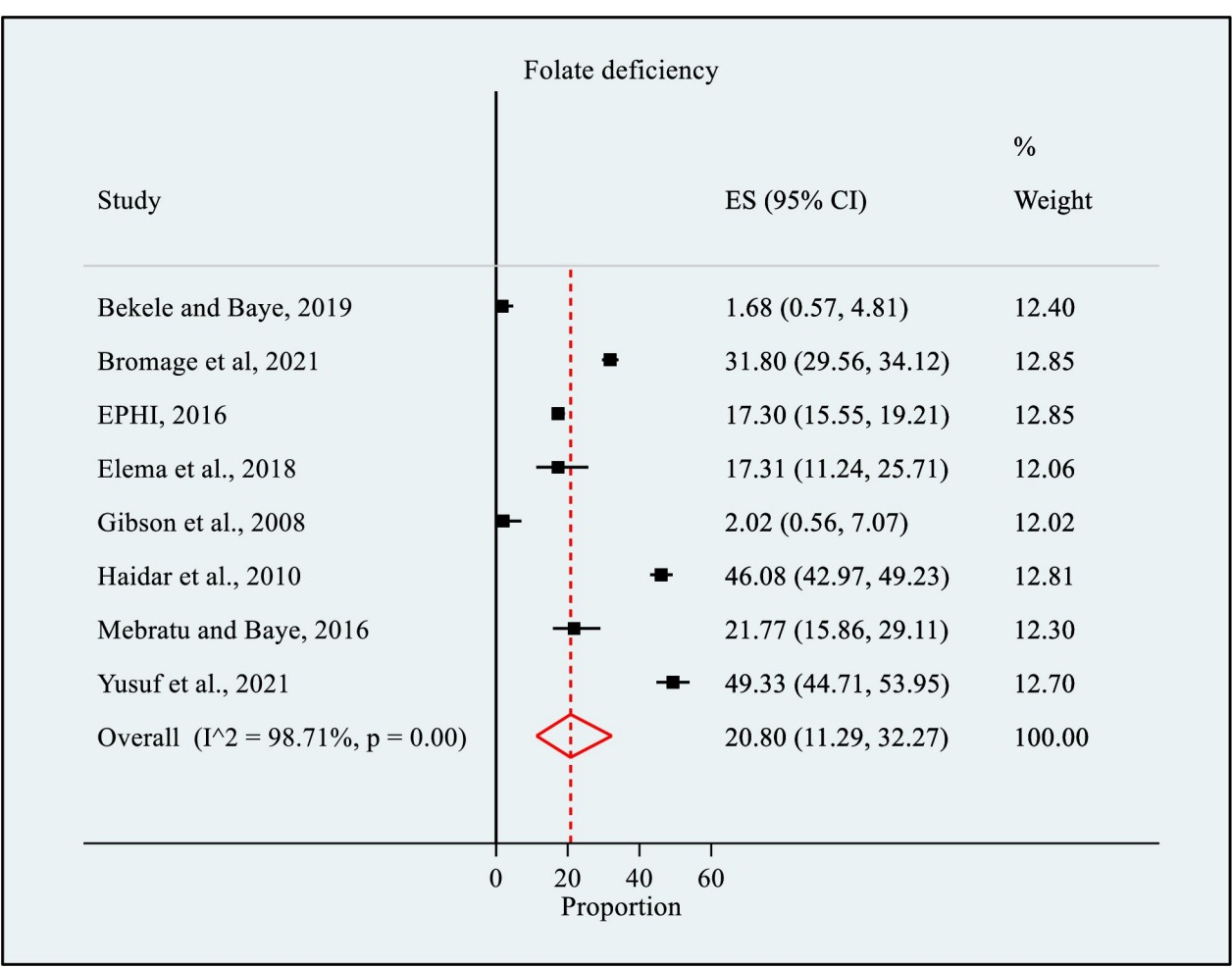

**Fig 3. Forest plot of the prevalence of folate deficiency among women of reproductive age in Ethiopia, 2022.**

## Subgroup analysis for mean serum/plasma folate concentration

To explore sources of heterogeneity, sub-group analyses were performed for the mean serum/plasma folate level by survey year and sampling technique. Consequently, the level of mean serum/plasma folate was higher among studies from 2015 or earlier (n = 3), (8.18 ng/ml, 95% CI: 5.38, 10.98) compared to those conducted after 2015 (n = 1) (4.50 ng/ml, 95% CI: 4.49, 4.51). There was considerable between-group heterogeneity (p = 0.01) (Fig 4).

Likewise, the mean serum/plasma folate was higher among studies which used non-probability sampling techniques (n = 2) (9.52 ng/ml, 95% CI: 5.70, 13.34) compared to those that used probability sampling (n = 2) (5.04 ng/ml, 95% CI: 3.97, 6.12). Significant heterogeneity was observed between the groups (p = 0.03) (Fig 5).

## Subgroup analysis for the prevalence of folate deficiency

We also performed subgroup analyses for the prevalence of FD to find out the possible sources of heterogeneity. Subgroup analysis by population/target group indicated that the prevalence of FD was higher among WRA (unspecified/general population group) (n = 2), (36.73%, 95%

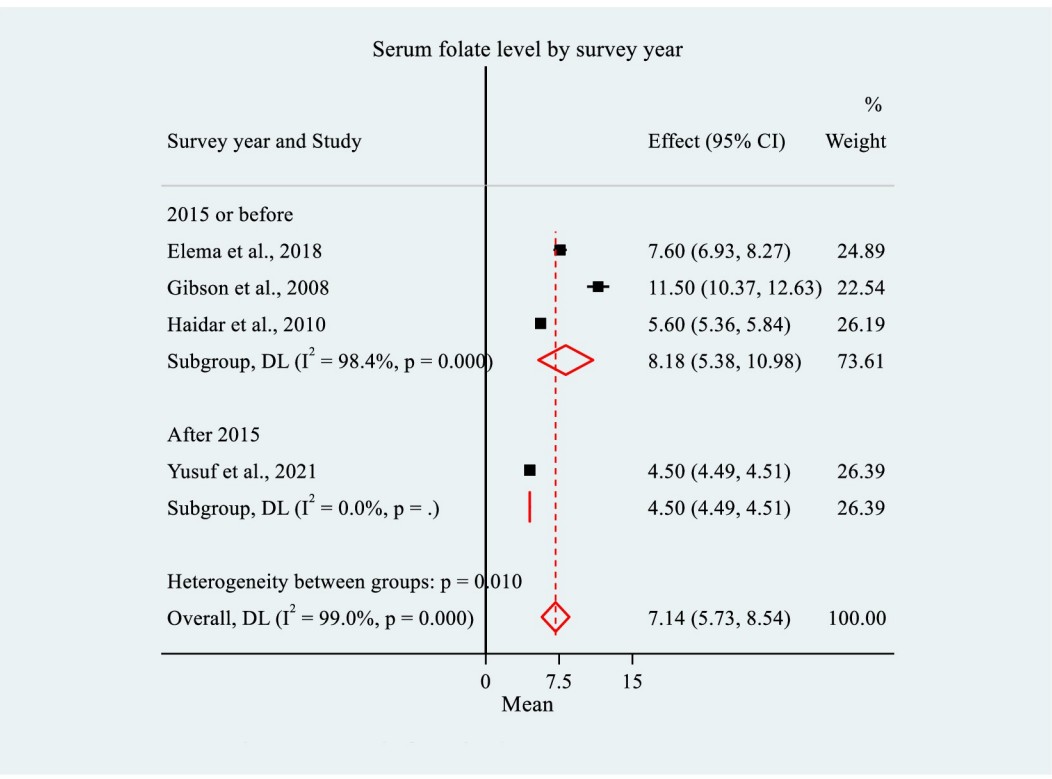

**Fig 4. Forest plot of subgroup analysis for mean serum/plasma folate level by survey year among women of reproductive age in Ethiopia, 2022.** NOTE: Weights and between-subgroup heterogeneity test are from random-effects model.

CI: 33.96, 39.55) than the non-pregnant/non-lactating (n = 2), (24.08%, 95% CI: 22.63, 25.57) and pregnant women (n = 4), (20.01, 95% CI: 3.34, 45.44). The between-groups heterogeneity was significant (p = 0.000) (Fig 6).

In the same way, the prevalence of FD was higher among women older than 30 years (n = 2), (36.73%, 95% CI: 33.96, 39.55) compared to those aged 30 years or less (n = 5), (19.53%, 95% CI: 7.10, 36.10). Significant between-groups heterogeneity was observed (p = 0.045) (Fig 7).

## Sensitivity analysis

We performed influential sensitivity (leave-one-out) analysis to assess the effect of each individual study on the observed heterogeneity. Accordingly, the overall point estimate fell within the CI of each sub-set of the studies obtained by leaving out exactly one study. This indicates that no individual study significantly affected the overall estimate of the rest of the studies (Table 2).

## Meta-regression

Meta-regression was performed for the mean serum/plasma folate level and FD for covariates to explain the possible sources of heterogeneities observed in the studies. Accordingly, sampling technique was found to be the only covariate significantly associated with mean serum/plasma folate. The application of non-probability sampling technique increased the mean

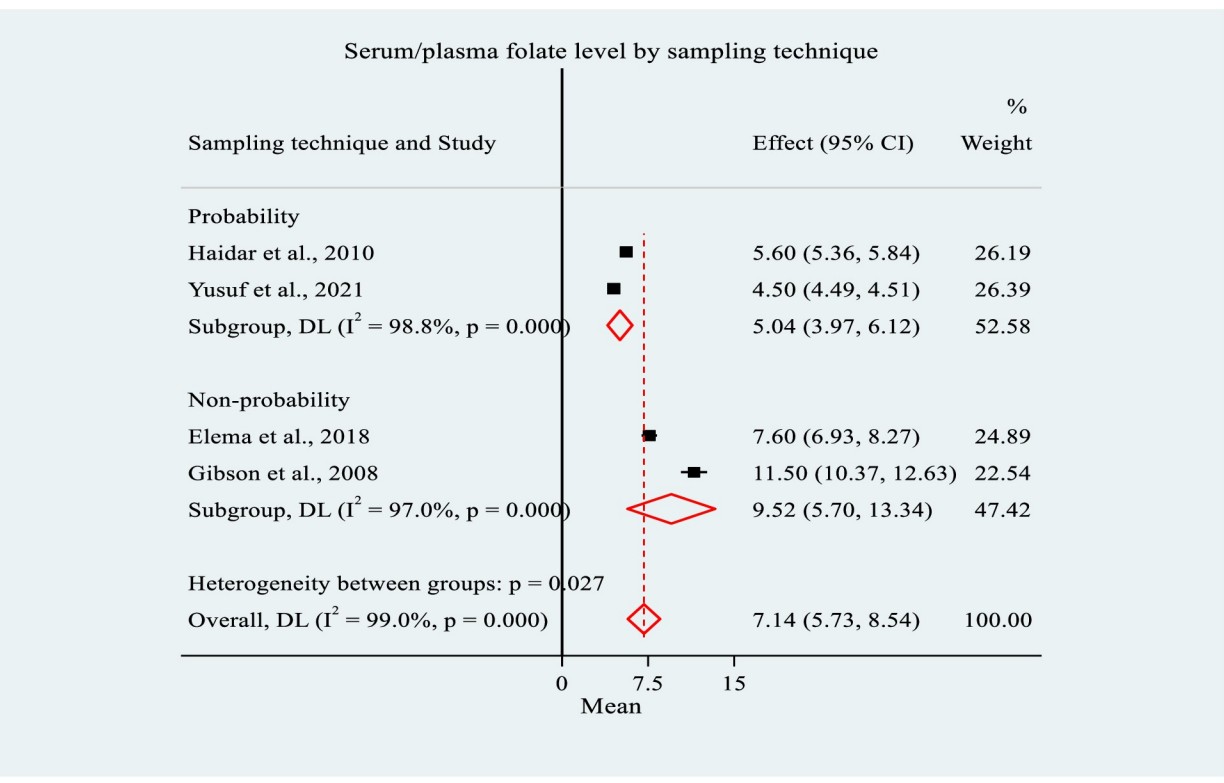

**Fig 5. Forest plot of subgroup analysis for mean serum/plasma folate level by sampling technique among women of reproductive age in Ethiopia, 2022.** NOTE: Weights and between-subgroup heterogeneity test are from random-effects model.

serum/plasma folate level by 4.45 ng/ml (95% CI: 0.57, 8.33; p = 0.025) compared to probability sampling method. No covariate showed a significant association with FD (Table 3).

## Discussion

Given that there is a paucity of evidence on folate status, particularly in low-income countries [22], this systematic review and meta-analysis was a step towards estimating the pooled mean serum/plasma folate and prevalence of FD among WRA—a vulnerable population group—in Ethiopia, a low-income country. This will provide an essential step for policymakers and other stakeholders in designing, planning and implementing future context-specific interventions on folate.

Ten studies were eligible for this review; however, two [10, 51] were not considered in the meta-analyses. One of these two studies reported folate insufficiency instead of FD [10], and the other reported both possible/marginal FD and FD combined into one [51]. On the other hand, only two studies reported factors associated with FD. We did not perform a meta-analysis because iron deficiency anemia was the single factor reported in these studies [4, 52]. Therefore, we calculated pooled mean serum/plasma folate and pooled prevalence of FD from four [4, 27, 29, 52] and eight [4, 27–32, 52] studies, respectively.

The current meta-analysis revealed that the pooled mean serum/plasma folate estimate was 7.14 ng/ml (95% CI: 5.73, 8.54). It was significantly higher among studies that applied non-probability sampling and those conducted in 2015 or earlier. Besides, the meta-regression

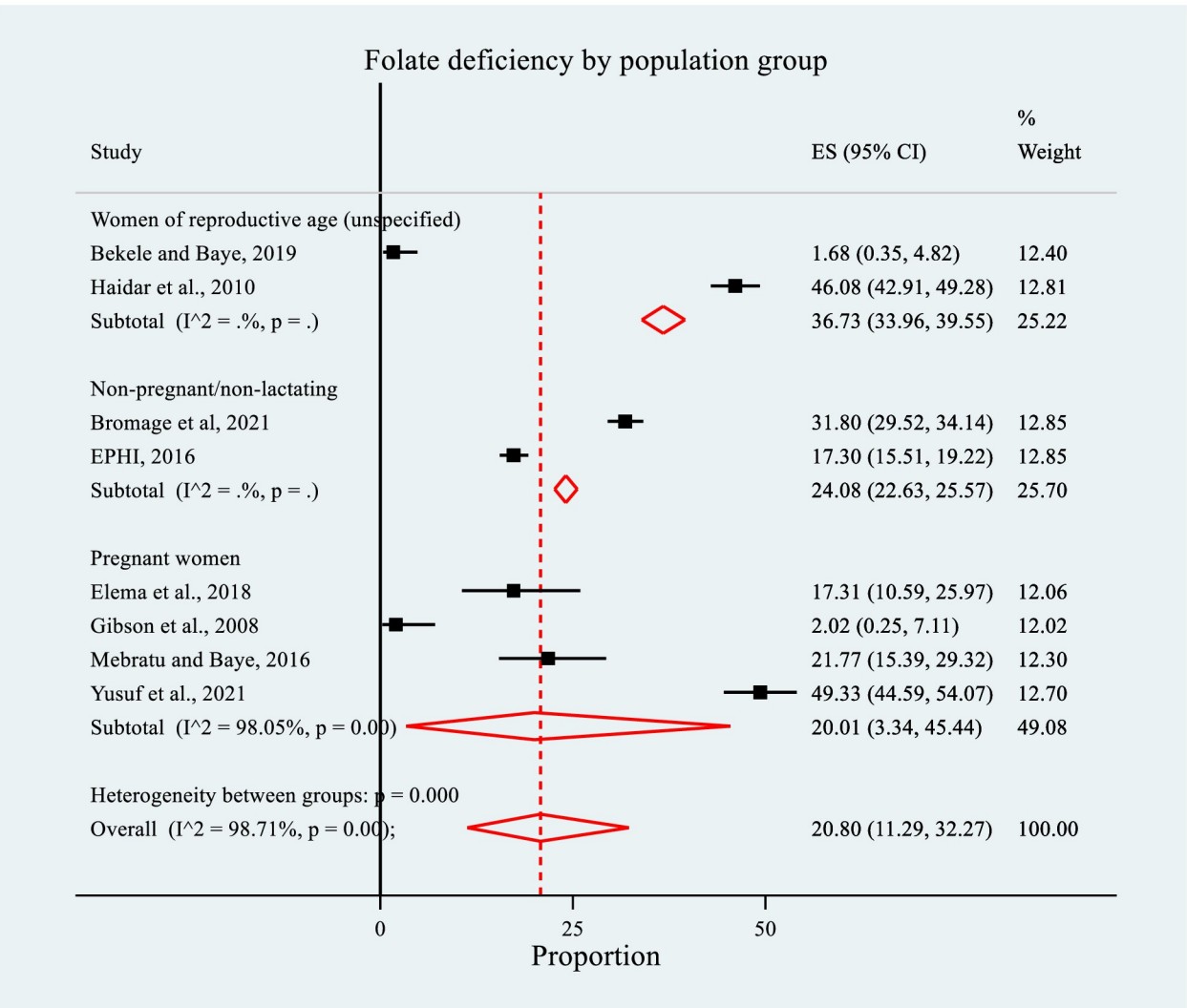

**Fig 6. Forest plot of subgroup analysis for the prevalence of folate deficiency by population group among women of reproductive age in Ethiopia, 2022.**

analysis showed that the non-probability sampling technique was significantly associated with the mean serum/plasma folate. Although there is no comparative meta-analysis, the mean serum/plasma folate was consistent with study findings from Nepal (5.9 ng/ml) [53], Georgia (7.2 ng/ml) [54], Iran (8.0 ng/ml) [55], Lebanon (8.4 ng/ml) [56] and Senegal (8.5 ng/ml) [24]. However, it was lower than the findings of various studies in different parts of the world, where the mean serum/plasma folate ranged from 8.6 ng/ml in Austria to 16.3 ng/ml in Ecuador [57–67]. On the other hand, it was slightly higher than one national study included in this meta-analysis (5.6 ng/ml) [4]. Additionally, it was higher than the findings of several studies, where the mean serum/plasma folate ranged from 2.3 ng/ml in Mongolia to 5.7 ng/ml in Turkey [23, 68–73].

The discrepancies in the mean serum/plasma folate could be due to the differences in socio-economic status, study scale, population group, study period, blood sample collection and storage method, the type of laboratory assay used, and eating habits. Our meta-analysis

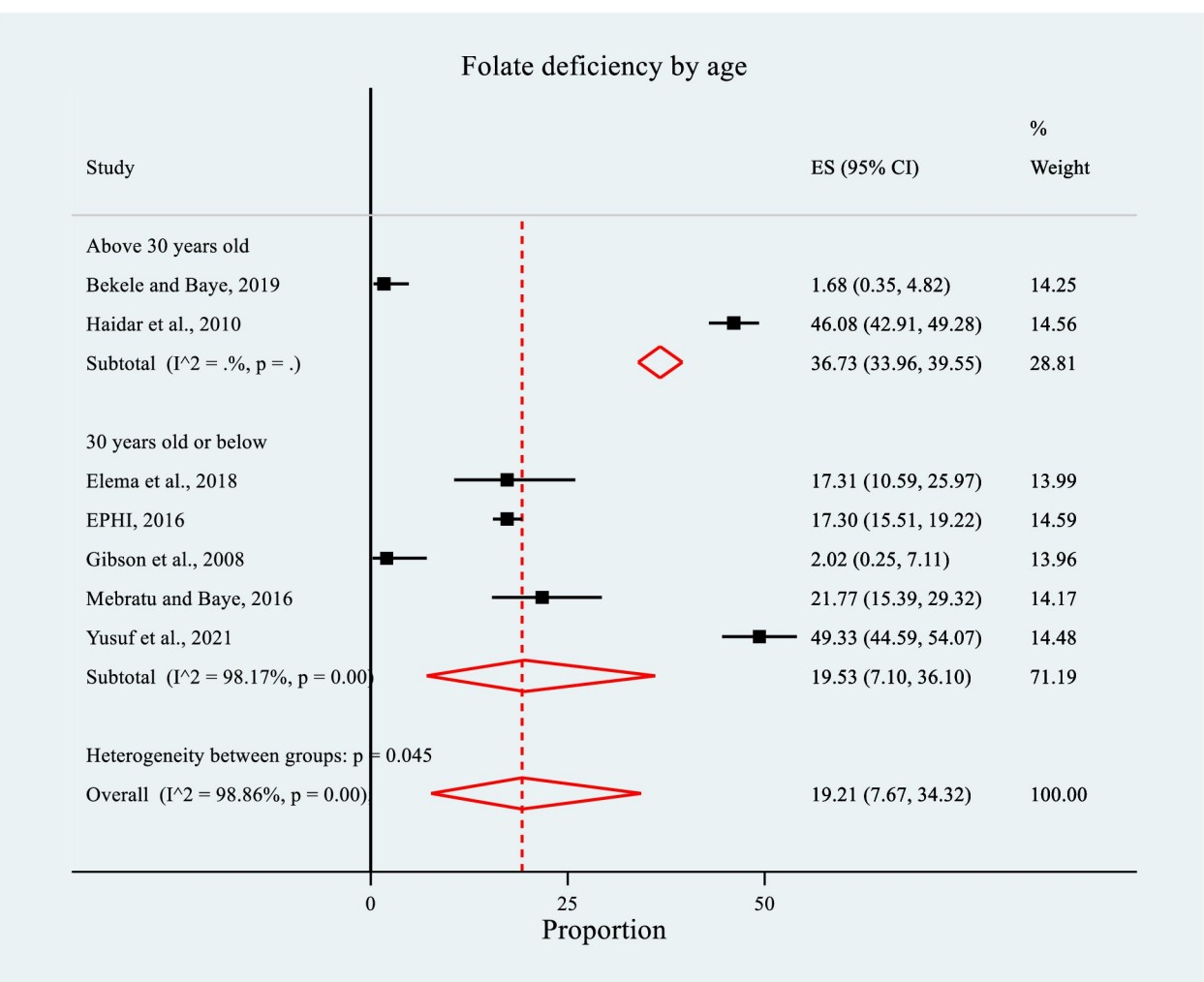

**Fig 7. Forest plot of subgroup analysis for the prevalence of folate deficiency by mean age among women of reproductive age in Ethiopia, 2022.**

**Table 2. Leave-one-out sensitivity analysis showing the influence of each individual study on the overall estimate in the mean serum/plasma folate level (n = 4) and the prevalence of folate deficiency (n = 8) among women of reproductive age in Ethiopia, 2022.**

| Study omitted | Pooled mean serum/plasma folate level in ng/ml (95% CI) | Pooled prevalence of folate deficiency (95% CI) |
| --- | --- | --- |
| Bekele and Baye, 2019 | — | 24.89 (14.93, 36.41) |
| Bromage et al, 2021 | — | 19.21 (7.67, 34.32) |
| EPHI, 2016 | — | 21.30 (10.58, 34.49) |
| Elema et al., 2018 | 6.94 (5.46, 8.43) | 21.29 (11.03, 33.77) |
| Gibson et al., 2008 | 5.84 (4.65, 7.03) | 24.49 (14.09, 36.66) |
| Haidar et al., 2010 | 7.83 (4.25, 11.41) | 17.64 (8.61, 28.98) |
| Mebratu and Baye, 20 | — | 20.65 (10.44, 33.20) |
| Yusuf et al., 2021 | 8.18 (5.38, 10.98) | 17.31 (8.30, 28.71) |
| **Combined (overall)** | **7.14 (5.73, 8.54)** | **20.80 (11.29, 32.27)** |

**Table 3. Meta-regression result of covariates to explain heterogeneity observed in the mean serum/plasma folate level (n = 4) and folate deficiency (n = 8) among women of reproductive age in Ethiopia, 2022.**

| Outcome variable | Covariates | Coefficient (95% CI) | Standard Error | P-value |
|---|---|---|---|---|
| | Survey year | | | |
| | 2015 or earlier | 3.70 (-3.01, 10.41) | 0.34 | 0.28 |
| | After 2015 | Reference | | |
| | Scale of study | | | |
| | National | -2.23 (-10.12, 5.65) | 4.02 | 0.58 |
| | Sub-national/local | Reference | | |
| | Study setting | | | |
| | Community-based | -0.43 (-8.90, 8.02) | 4.31 | 0.92 |
| | Institution-based | Reference | | |
| | Sampling technique | | | |
| | Non-probability | **4.45 (0.57, 8.33)** | **1.98** | **0.03** |
| | Probability | Reference | | |
| | Sample size | | | |
| | <500 | 2.23 (-5.65, 10.12) | 4.02 | 0.58 |
| | >500 | Reference | | |
| | Mean age | -0.09 (-1.28, 1.10) | 0.61 | 0.89 |
| **Folate deficiency** | Survey year | | | |
| | 2015 or earlier | -0.07 (-0.34, 0.21) | 0.14 | 0.64 |
| | After 2015 | Reference | | |
| | Scale of study | | | |
| | National | 0.12 (-0.14, 0.38) | 0.13 | 0.34 |
| | Sub-national/local | Reference | | |
| | Study setting | | | |
| | Community-based | 0.06 (-0.25, 0.38) | 0.16 | 0.70 |
| | Institution-based | Reference | | |
| | Population group | | | |
| | Non-pregnant/non-lactating | -0.01 (-0.41, 0.41) | 0.21 | 0.98 |
| | Pregnant | -0.01 (-0.38, 0.35) | 0.19 | 0.94 |
| | WRA (unspecified) | Reference | | |
| | Sampling technique | | | |
| | Non-probability | 0.16 (-0.42, 0.10) | 0.13 | 0.23 |
| | Probability | Reference | | |
| | Laboratory assay | | | |
| | Protein binding | 0.19 (-0.11, 0.49) | 0.15 | 0.21 |
| | Microbiological | Reference | | |
| | Sample size | | | |
| | <500 | -0.12 (-0.38, 0.14) | 0.13 | 0.34 |
| | >500 | Reference | | |
| | Mean age | -0.01 (-0.52, 0.04) | 0.02 | 0.85 |

focused on one of the low income-countries (i.e., Ethiopia) with limited coverage and adherence to folic acid supplementation. Although mandatory folic acid fortification has been endorsed by the country [45], its translation into action has been limited. Contrary to our review, most other studies were primary studies (not meta-analyses) from middle- and high-income countries. Additionally, these middle- and high-income countries have better coverage

and adherence to folic acid supplementation and long history of endorsing of mandatory folic acid fortification and effective implementation.

The other point is that different studies use different methods and procedures for blood sample collection and storage and laboratory analysis, such as laboratory assays. Folate results show poor comparability across the laboratory methods, sometimes even within the same analytical technique [74, 75]. A single assay with different calibrators and laboratory set-ups can even produce different results [22], making it challenging to compare folate concentrations across surveys.

Differences in dietary habits across communities, including fasting and folic acid supplementations, could also explain the discrepancies, as serum/plasma folate is easily affected by recent intake [76]. Serum folate concentrations are approximately 10% higher in nonfasting than in fasting persons, complicating sample collection in field studies [77].

A significant driver of much of the recent public health attention to the importance of folate has been the link to NTDs [16]. In light of this, our systematic review and meta-analysis highlighted that the pooled prevalence of FD was 20.80% (95% CI: 11.29, 32.27), which was four times higher than the threshold for a country-wide public health problem (i.e., greater than 5%) [16–19]. Therefore, FD is clearly a concern in Ethiopia. Furthermore, significant differences were observed in the sub-group analysis by population group and mean age of WRA; meaning the prevalence was higher among WRA (unclassified population group) than the non-pregnant/non-lactating and pregnant women. In addition, it was higher among women above 30 years. However, no covariate was significantly associated with FD.

The prevalence of FD in this meta-analysis was consistent with one systematic review and meta-analysis, which pooled 45 surveys conducted in 39 countries from 2000 to 2014. Only 11 surveys were from lower economic countries; the pooled FD was greater than 20% for most countries [22]. Likewise, it was consistent with a national survey report (17.3%) included in our meta-analysis. However, the prevalence was higher than two study findings reported from India, with an FD prevalence of 3.2% and 3.5% [60, 73] and another study from Belize (4.1%) [9]. On the contrary, the prevalence of FD was lower in our meta-analysis than in one primary study done in Senegal (54.8%) [24].

The differences in socio-economic and demographic characteristics, study scale, setting, period and population, laboratory methods, cut-offs, and feeding habits could explain the variations in the prevalence of FD. The three studies [9, 60, 73] with inconsistent findings with our meta-analysis were single (not meta-analyses). This could be explained by the fact that a single primary study may not be comparable with a meta-analysis, which results from statistical analysis of several studies. The other reason for the difference could be the differences in the study population. For example, the Indian studies focused only on pregnant women. In most settings, pregnant women have more access to folic acid supplementation than other sub-groups of WRA. On the other hand, Belize is an upper middle-income country, which adopted the mandatory folic acid fortification into wheat flour in 1998 [78], whereas Ethiopia adopted it in 2022 [45]. This could be why the prevalence of FD is much lower in Belize than in our meta-analysis. The prevalence of FD in the Senegalese study was far higher than in our meta-analysis. The target group in the Senegalese study was WRA, where 90.8% were not pregnant [24]. This means most do not have access to folic acid supplementation provided during pregnancy as part of antenatal care.

Most importantly, the differences in FD could be due to variations in laboratory assays, cut-offs, and dietary practices [22, 76]. Folate is not stable and is found in the body in numerous forms. Therefore the various assays have a different affinity to the different active forms of folate [22]. Furthermore, studies use different cut-off points depending on the type of indicator used to define FD. In our meta-analysis, the included studies used two sets of cut-offs,

published by the WHO [47]. Therefore, folate status should be measured using a microbiological assay as recommended by the WHO that is consistent with regard to common reagents and protocols along with assay-matched cut-offs to define FD and insufficiency for appropriate comparisons across studies [47, 79].

The systematic review and meta-analysis findings can be used as input for policies and programs and a gateway for further epidemiological and nutritional studies and meta-analyses. However, the systematic review and meta-analysis has some limitations. For example we could not produce the pooled estimates for folate insufficiency (an indicator of the risk of NTDs) and factors associated with FD or insufficiency due to the scarcity of adequate primary studies. Additionally, we could not estimate the pooled mean or prevalence of FD using RBC folate, which reflects long-term folate status, because few studies used this biomarker. Moreover, using different laboratory assays may have under- or over-estimated the results.

## Conclusions

The systematic review and meta-analysis highlighted that the pooled mean serum/plasma folate concentration among WRA was relatively low. Significant differences were observed in mean serum/plasma folate concentration depending on the sampling techniques used by the studies and the study period (survey year). The pooled prevalence of FD was reported in one-fifth of the WRA, indicating its public health significance in Ethiopia. There was a significant difference in the prevalence by population group and mean age of WRA.

Therefore, the Ethiopian health and nutrition policies should design and strengthen public health strategies to address FD among WRA. Since the magnitude of FD varies throughout the country, targeted interventions are required to address the deficiency. Public health messages should be disseminated through different platforms to improve women's knowledge of folate, its deficiency, prevention strategies, and folate-rich foods. The availability of folate-rich foods should be ensured by improving production, processing, preservation, pricing, and marketing. Additionally, the consumption of folate-rich foods and folic acid supplementation with a particular focus on coverage and adherence should be strengthened. Preconception folic acid supplementation should also be considered to reduce NTDs. Folic acid fortification is considered one of the most successful public health initiatives to reduce NTDs and has been an effective intervention strategy in many countries. Thus, the government of Ethiopia, in collaboration with partners, should invest more in folic acid fortification. The mandatory folic acid fortification endorsed by the country should be transformed into action.

## Supporting information

**S1 Dataset. The dataset from which the results of the study were produced (STATA file).**
(DTA)

**S1 Table. Risk of bias and quality assessment result of the included 10 studies using the Joanna Briggs Institute (JBI) Critical Appraisal Checklist for Studies Reporting Prevalence Data, 2022.**
(DOCX)

**S2 Table. PRISMA 2020 checklist.**
(DOCX)

**S1 Text. Full search strategy of studies.**
(DOCX)

**S2 Text. A copy of manuscript with track changes (copyedited for language errors).**
(DOCX)

## Acknowledgments

The authors are grateful to Dr. Tara D'Ann Wilfong for her contribution in copyediting our manuscript for language usage, spelling and grammar.

## Author Contributions

**Conceptualization:** Berhe Gebremichael, Hirbo Shore Roba, Alemeshet Getachew, Dejene Tesfaye, Haftu Asmerom.

**Data curation:** Berhe Gebremichael, Dejene Tesfaye, Haftu Asmerom.

**Formal analysis:** Berhe Gebremichael, Hirbo Shore Roba, Alemeshet Getachew, Dejene Tesfaye, Haftu Asmerom.

**Investigation:** Berhe Gebremichael.

**Methodology:** Berhe Gebremichael, Hirbo Shore Roba, Alemeshet Getachew, Dejene Tesfaye, Haftu Asmerom.

**Project administration:** Berhe Gebremichael.

**Resources:** Berhe Gebremichael.

**Software:** Berhe Gebremichael, Hirbo Shore Roba, Alemeshet Getachew, Dejene Tesfaye, Haftu Asmerom.

**Supervision:** Berhe Gebremichael, Alemeshet Getachew.

**Validation:** Berhe Gebremichael, Hirbo Shore Roba, Alemeshet Getachew.

**Visualization:** Berhe Gebremichael, Hirbo Shore Roba, Alemeshet Getachew.

**Writing – original draft:** Berhe Gebremichael.

**Writing – review & editing:** Berhe Gebremichael, Hirbo Shore Roba, Alemeshet Getachew, Dejene Tesfaye, Haftu Asmerom.

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
