## [Decision Letter · Decision Letter 0]

13 Feb 2023

PONE-D-22-30184Folate deficiency among women of reproductive age in Ethiopia: a systematic review and meta-analysisPLOS ONE

Dear Dr. Berhe,

Thank you for submitting your manuscript to PLOS ONE. After careful consideration, we feel that it has merit but does not fully meet PLOS ONE’s publication criteria as it currently stands. Therefore, we invite you to submit a revised version of the manuscript that addresses the points raised during the review process.

ACADEMIC EDITOR:Details comments are placed below for your action. Please kindly address all the major comments raised by the reviewers and the editor while submittig your manuscript. The manuscript has many major methodological flaws which need to be addressed while resubmission.

We look forward to receiving your revised manuscript.

Kind regards,

Abdu Oumer

Academic Editor

PLOS ONE

Journal Requirements:

Additional Editor Comments:

Dear authors,

After a period of initial checks and thorough peer review, we come up with the review result. The manuscript has a merit for publication but it has many technical and other major issues which need to be addressed during the revision. Once, you addressed the requested suggestions at satisfactory level, decision on the manuscript will be made.

Thanks again for choosing us.

Kind regards,

Academic editor

Reviewers' comments:

Reviewer's Responses to Questions

**Comments to the Author**

1. Is the manuscript technically sound, and do the data support the conclusions?

Reviewer #1: Yes

Reviewer #2: Yes

2. Has the statistical analysis been performed appropriately and rigorously? 

Reviewer #1: Yes

Reviewer #2: Yes

3. Have the authors made all data underlying the findings in their manuscript fully available?

Reviewer #1: Yes

Reviewer #2: No

4. Is the manuscript presented in an intelligible fashion and written in standard English?

Reviewer #1: Yes

Reviewer #2: Yes

5. Review Comments to the Author

Reviewer #1: Dear Plos ONE team of editorials, thank you for the chance given to me to review a manuscript titled “Folate deficiency among women of reproductive age in Ethiopia: a systematic review and meta-analysis”. One third of global women experience iron deficiency anemia and the study will have a paramount importance for the design and implementation of appropriate public health interventions and will fill the gap of literature on the field. I should also appreciate that the study is well referenced. The following are my comments;

1. The criteria for the inclusion and exclusion of the articles should be refined and clear cut. The identified article for assessing (a) serum concentration (b) for assessing folate deficiency should go together

2. Have you included only Pregnant women or non-pregnant women or both? How many are institutional and how many are community based?

3. Are you studying for clinical or epidemiological significance or both or?

4. When do we say folate deficiency is called “major public health problem” globally and in Ethiopia?

5. In case when there is ambiguity between the two independent extractors who had decided?

6. Where are the other search engines like other Universities repository and Addis Ababa University and EPHA joint publication Ethiopian Journal of Health development and Jimma University’s Journal of Health Sciences…etc

7. Where is the sub-group analysis based on the age, residence, education status of the participants

8. The result and the discussion contents should contain what it tends to contain.

9. The recommendations are already under implementation. Hence, do you have other innovative recommendations drawn from your findings?

10. The language and the statistics need mainly refinement. The language needs major revision

Reviewer #2: It’s a well-written and interesting study.

Abstract

1. Include the full names of the following abbreviated databases, AJOL, VMNIS, GHDx

2. Right the full names of WHO

3. The recommendation should consider planned and implemented innervations aimed to reduce folate deficiency. The interventions you suggested are already within the government's focus. However, there are shortcomings in the coverage, translating fortification policies into actions. The recommendation should be rewritten in light of this.

4.

Introduction

1. The introduction should clearly state why folate deficiency has been considered a public health problem, especially among women of reproductive age.

2. Mention what are the currently implemented (iron-folic acid supplementation) and planned interventions (fortification).

3. The last paragraph should be focused on folate deficiency only.

4. You have stated that “However, evidence on the prevalence of FD among WRA are not strong enough to the level policymakers could develop evidence-based intervention strategies in the country except for folic acid supplementation for pregnant women who visit health facilities for antenatal care follow-up.” However, the Ethiopian Standard Council endorsed the mandatory fortification of edible oil and wheat flour on 10th June 2022, with the currently available evidence. This seems to be in contradiction with your claim. In light of this rewrite the above-mentioned section. You can find this information through this link https://www.nutritionintl.org/news/all-news/government-endorses-mandatory-food-fortification-prevent-high-burden-neural-tube-defects-ethiopia

Method

Search strategy

1. It has been mentioned that Addis Ababa university instructional repositories have been used. This is a great approach to identifying unpublished studies. Makes sure that the thesis or dissertation you have identified has not been published and cross-check for duplication.

2. Other major national universities' institutional repositories and research institution repositories should also be included in the search strategy.

Eligibility criteria

3. Mention how studies from the same survey have been handled

4. In line 107 remove “NOT”, since you haven’t used this boolean operator in the search strategy

Result

5. in line 175 remove “Id”

6. summarise characteristics of studies and participants in table format

7. correct table 1, has been split into two places

8. Table 1 has been cited on line number 225, however, this paragraph is not related to the findings reported in table 1.

9. Throughout the manuscript replace “participants” with woman of reproductive age (WRA)

10. Table 1 has been cited on line number 277, however, this paragraph is not related to the findings reported in table 1

11. Perform a sub-group analysis based on the laboratory method used to asses folate concentration If adequate studies can be found within the subgroups.

12. Please include types of laboratory methods used to analyse folate concentration as a covariate in the meta-regression.

13. Discuss further the similarity and difference in mean serum folate concentration and folate deficiency with studies from other nations; explain the similarity and difference from the perspective of implemented interventions aimed at reducing folate deficiency.

14. From lines 391 to 393 you have stated that “All the included studies into the meta-analysis were cross-sectional, and temporal relationship could not be established between the covariates and the outcome variables”. Remove these statements as the aim of this systematic review and meta-analysis is not to establish causality or explore folate deficiency determinants.

15. You have mentioned that different laboratory assays are used. Discuss how the difference in laboratory assays used may affect your finding, and comparability of studies.

Conclusion

16. Rewrite the concussion making it clear and concise

17. In line 403 replace “women” with women of reproductive age

18. From lines 403-404 you have stated that “Additionally, the pooled prevalence of FD was reported in one-fifth of the women, indicating its public health significance in Ethiopia.”. It is also possible to justify why folate deficiency is public health problem from the perspective of neural tube defect

19. You have recommended multiple interventions targeting folate deficiency among these interventions fortification has been proven to be the most effective one. On your recommendation, more emphasis should be given to fortification interventions.

6. PLOS authors have the option to publish the peer review history of their article (what does this mean?). If published, this will include your full peer review and any attached files.

Reviewer #1: No

Reviewer #2: No

---

## [Author Response · Author response to Decision Letter 0]

3 Apr 2023

1. Comments from the academic editor and responses

Comment-1: Please ensure that your manuscript meets PLOS ONE's style requirements, including those for file naming.

Response: The revised manuscript has been modified to PLOS ONE's style

Comment-2: We suggest you thoroughly copyedit your manuscript for language usage, spelling, and grammar. Upon resubmission, please provide the following: the name of the colleague or the details of the professional service that edited your manuscript, a copy of your manuscript showing your changes by either highlighting them or using track changes (uploaded as a *supporting information* file), and a clean copy of the edited manuscript (uploaded as the new *manuscript* file)”

Response: We have made our manuscript be copyedited by a native American English speaker—Dr. Tara Wilfong (MD, MPH). She is an Associate Professor at Haramaya University. We indicated this in the acknowledgement section of the manuscript 

Comment-3: Please include a separate caption for each figure in your manuscript.

Response: We accepted the comment, and we corrected it

Comment-4: Please include captions for your Supporting Information files at the end of your manuscript, and update any in-text citations to match accordingly.

Response: The comment is accepted and corrected as per the request

2. Comments of reviewer #1 and responses

Comment-1: The criteria for the inclusion and exclusion of the articles should be refined and clear cut.

Response: We appreciated the comment and we modified it as per the recommendation

Comment-2: The identified article for assessing (a) serum concentration (b) for assessing folate deficiency should go together.

Response: The number of articles for serum concentration and folate deficiency are not the same because some studies report only mean serum folate, some report only folate deficiency, and others report both. That’s why the number of identified articles are not the same for the serum folate and folate deficiency. 

Comment-3: Have you included only Pregnant women or non-pregnant women or both? How many are institutional and how many are community based?

Response: We included women of reproductive age in general (both pregnant and non-pregnant women). In the identified articles, some studies were done among pregnant women, some in non-pregnant and others generally in women of reproductive age without specifying as pregnant or non-pregnat. Regarding the study setting, six of the studies were community based, while four studies were institution based. These all are clearly indicated on page 9 of the manuscript.

Comment-4: Are you studying for clinical or epidemiological significance or both or?

Response: The significance of this study is mainy a baseline framework for epidemiological and nutritional researches. It could also be used by policy makers/programmers as an input in planning prevention programs against folate deficiency. This is indiated on page 21, the last paragraph of the discussion

Comment-5: When do we say folate deficiency is called “major public health problem” globally and in Ethiopia?

Response: To the best of our knowledge, clear threshold based on the prevalence of folate deficiency has not been established. This is due to the use of different analytical methods and different biomarker cutoff points, and lack population-based relevant data. However, it is considered as a public health problem because it results in serious consequences such as megaloblastic anemia and neural tube defects. Generally, prevalence of folate deficiency >5% is represents a threshold for country-wide public health problem (https://www.ncbi.nlm.nih.gov/pmc/articles/PMC4478945/ and https://www.tandfonline.com/doi/abs/10.1080/16070658.2010.11734327). We revised the introduction part accordingly.

Comment-6: In case when there is ambiguity between the two independent extractors who had decided?

Response: It was resolved through discussions and consultations with a third author. This is indicated on page 7.

Comment-7: Where are the other search engines like other Universities repository and Addis Ababa University and EPHA joint publication Ethiopian Journal of Health development and Jimma University’s Journal of Health Sciences…etc

Response: Thank you for the comment. We tried to search the institutional repositories of universities including Addis Ababa University, Jimma University, Hawassa University, Haramaya University, Arbaminch University, University of Gondar, Bahir Dar University and Mekelle University. However, we could not get theses and dissertations on folate deficiency, except Addis Ababa University. Even the repositories of some universities were not active during the searching period (e.g. Haramaya University). Most of the theses we got from the other universities were on iron-folic acid supplementation adherence. This is why we included only Addis Ababa University in the review report. Now, we revised it as we also did the search for the other Universities.

Concerning the journals such as Ethiopian Journal of Health Development and Journal of Health Sciences, we do not search on each and every journal. Rather, we search on databases or platforms like PubMed, Google scholar, Embase, CINAHL, given that the journals are expected to be indexed or their articles are at least available online. There are hundreds of thousands of journals, if not millions, in the world. So, searching all these journals individually is difficult. What is recommended is to search on the indexing platforms/databases. 

Comment-8: Where is the sub-group analysis based on the age, residence, education status of the participants?

Response: The sub-group analysis for mean age is indicated on page 16-17 (table 2). However, we did not do sub-group analysis for residence and education. Regarding residence, most of the studies did not report the variable (except one study). In addition, some of the studies were conducted in rural areas only and some were done only in urban areas. Others seem as they included urban and rural settings, but they did not report the variable residence in their result section. And we could not extract data that could be pooled up. We did not also extract data for the variable education. Although two studies did not report results for educational status, the issue is different from residence. Almost all of the studies used different categories for the variable “educational status.” Therefore, there were inconsistencies in the categories/responses among the studies. E.g. one study reported it as ‘illiterate and literate’, another study reported it as ‘formal education and no formal education’, other study reported it as ‘educated, read/write, and illiterate’, another study also reported as ‘cannot read/write, informal education and formal education’ and so on. It was difficult to pool up these different categories into one. E.g some of those under the category ‘no formal education’ could include those who are able to read/write from informal education (e.g. religious education) and illiterate ones 

Comment-9: The result and the discussion contents should contain what it tends to contain.

Response: We accepted the comment and tried to revise accordingly

Comment-10: The recommendations are already under implementation. Hence, do you have other innovative recommendations drawn from your findings?

Response: We appreciated the comment and we revised it. We recommended the programs and strategies under implementation to be strengthened because there are problems in implementation. E.g. there is problem in folic acid supplementation coverage and adherence. There is also shortcoming in translating the fortification program into action. We have also recommended some additional interventions. 

Comment-11: The language and the statistics need mainly refinement. The language needs major revision

Response: We revised for any inconsistencies and discrepancies in the document, including the statistics. Additionally, a native English speaker edited the document for language and editorial issues.

3. Comments of reviewer #2

Abstract

Comment-1: Include the full names of the following abbreviations: AJOL, VMNIS, GHDx, WHO

Response: We accepted the comment and corrected it accordingly

Comment-2: The recommendation should consider planned and implemented innervations aimed to reduce folate deficiency. The interventions you suggested are already within the government's focus. However, there are shortcomings in the coverage, translating fortification policies into actions. The recommendation should be rewritten in light of this.

Response: We revised it according to the advice

Introduction

Comment-1: The introduction should clearly state why folate deficiency has been considered a public

Response: We accepted your comment and revised it accordingly

Comment-2: Mention what are the currently implemented (iron-folic acid supplementation) and planned interventions (fortification).

Response: We revised it as per the comment

Comment-3: The last paragraph should be focused on folate deficiency only.

Response: Corrected accordingly

Comment-4: You have stated that “However, evidence on the prevalence of FD among WRA are not strong enough to the level policymakers could develop evidence-based intervention strategies in the country except for folic acid supplementation for pregnant women who visit health facilities for antenatal care follow-up.” However, the Ethiopian Standard Council endorsed the mandatory fortification of edible oil and wheat flour on 10th June 2022, with the currently available evidence. This seems to be in contradiction with your claim. In light of this rewrite the above-mentioned section.

Response: Thank you for the source, and we have re-written it as per your recommendation

Methods

Comment-1: It has been mentioned that Addis Ababa university instructional repositories have been used. This is a great approach to identifying unpublished studies. Makes sure that the thesis or dissertation you have identified has not been published and cross-check for duplication.

Response: We appreciate your concern, and we actually faced with such issues. One Masters thesis was found published, but we excluded the thesis and included the published one

Comment-2: Other major national universities' institutional repositories and research institution repositories should also be included in the search strategy.

Response: The comment is appreciated. We tried to search the institutional repositories of universities including Addis Ababa University, Jimma University, Hawassa University, Haramaya University, Arbaminch University, University of Gondar, Bahir Dar University and Mekelle University. However, we could not get theses and dissertations on folate deficiency, except Addis Ababa University. Even the repositories of some universities were not active during the searching period (e.g. Haramaya University). Most of the theses we got from the other universities were on iron-folic acid supplementation adherence. This is why we included only Addis Ababa University in the review report. Now, we have revised it as we also did the search for the other Universities. Additionally, we considered the institutional repositories of the research centers/institues such as the Ethiopian Public Health Institute (EPHI), Ethiopian Health and Nutrition Research Institute (EHNRI) and Ethiopian Nutrition Institute (ENI). However, we obtained only one study from the EPHI (i.e. Ethiopia National Micronutrient Survey, 2016), one of the studies included in the meta-analysis. This study was also found in the Global Health Data Exchange (GHDx), which has been mentioned in the manuscript as a separate database. We have revised our manuscript to indicate as we searched the websites and repositories of these research institutes.

Comment-3: Mention how studies from the same survey have been handled in the eligibility criteria

Response: We indicated in the respective section (page 6-7)

Comment-4: 4. In line 107 remove “NOT”, since you haven’t used this boolean operator in the search strategy

Response: We removed it

Results

Comment-1: In line 175 remove “Id”

Response: We removed it

Comment-2: Summarise characteristics of studies and participants in table format

Response: We summarized it in table form and labeled it ‘Table 1’

Comment-3: Correct table 1, has been split into two places

Response: Corrected

Comment-4: Table 1 has been cited on line number 225, however, this paragraph is not related to the findings reported in table 1.

Response: We appreciated for this critical comment and we corrected all the inconsistencies in the manuscript, including table 1. In the revised manuscript, this table is named ‘Table 2’

Comment-5: Throughout the manuscript replace “participants” with woman of reproductive age (WRA)

Response: Corrected

Comment-6: Table 1 has been cited on line number 277, however, this paragraph is not related to the findings reported in table 1

Response: We have corrected it

Comment-7: Perform a sub-group analysis based on the laboratory method used to asses folate concentration if adequate studies can be found within the subgroups

Response: The reason why we did not report on the sub-group analysis by laboratory method was due to inadequate studies for the microbiological assay (MBA) method.

Comment-8: Please include types of laboratory methods used to analyse folate concentration as a covariate in the meta-regression.

Response: We performed the analysis and included it as a covariate, but no significant association shown (table 3, page 20-21)

Comment-9: Discuss further the similarity and difference in mean serum folate concentration and folate deficiency with studies from other nations; explain the similarity and difference from the perspective of implemented interventions aimed at reducing folate deficiency.

Response: We tried to revise it accordingly

Comment-10: From lines 391 to 393 you have stated that “All the included studies into the meta-analysis were cross-sectional, and temporal relationship could not be established between the covariates and the outcome variables”. Remove these statements as the aim of this systematic review and meta-analysis is not to establish causality or explore folate deficiency determinants.

Response: We removed it 

Comment-11: You have mentioned that different laboratory assays are used. Discuss how the difference in laboratory assays used may affect your finding, and comparability of studies.

Response: We have discussed it in bothe the mean and the prevalence

Conclusions

Comment-1: Rewrite the concussion making it clear and concise 

Response: We did some modifications to make it clear

Comment-2: In line 403 replace “women” with women of reproductive age

Response: We have replaced it accordingly

Comment-3: From lines 403-404 you have stated that “Additionally, the pooled prevalence of FD was reported in one-fifth of the women, indicating its public health significance in Ethiopia.”. It is also possible to justify why folate deficiency is public health problem from the perspective of neural tube defect

Response: It could be justified, but as you said above, the main aim of the study is to assess the prevalence of folate deficiency. So, the conclusion should also be in line with the objective. We have already shown the public health significance of FD using the burden of NTDs in the introduction and discussion parts

Comment-4: You have recommended multiple interventions targeting folate deficiency among these interventions fortification has been proven to be the most effective one. On your recommendation, more emphasis should be given to fortification interventions.

Response: We have revised the recommendation based on the comment

End===============

---

## [Editor Report · Decision Letter 1]

19 Apr 2023

Folate deficiency among women of reproductive age in Ethiopia: a systematic review and meta-analysis

PONE-D-22-30184R1

Dear Dr. Gebremichael,

We’re pleased to inform you that your manuscript has been judged scientifically suitable for publication and will be formally accepted for publication once it meets all outstanding technical requirements.

Kind regards,

Abdu Oumer

Academic Editor

PLOS ONE
---

## [Editor Report · Acceptance letter]

25 Apr 2023

PONE-D-22-30184R1 

Folate deficiency among women of reproductive age in Ethiopia: a systematic review and meta-analysis 

Dear Dr. Gebremichael:

I'm pleased to inform you that your manuscript has been deemed suitable for publication in PLOS ONE. Congratulations! Your manuscript is now with our production department. 

Kind regards, 

on behalf of

Dr. Abdu Oumer 

Academic Editor

PLOS ONE